# Predicting Analyte Concentrations from Electrochemical Aptasensor Signals Using LSTM Recurrent Networks

**DOI:** 10.3390/bioengineering9100529

**Published:** 2022-10-06

**Authors:** Fatemeh Esmaeili, Erica Cassie, Hong Phan T. Nguyen, Natalie O. V. Plank, Charles P. Unsworth, Alan Wang

**Affiliations:** 1Department of Engineering Science, University of Auckland, Auckland 1010, New Zealand; 2School of Chemical and Physical Sciences, Victoria University of Wellington, Wellington 6021, New Zealand; 3The MacDiarmid Institute for Advanced Materials and Nanotechnology, Victoria University of Wellington, Wellington 6021, New Zealand; 4Auckland Bioengineering Institute, University of Auckland, Auckland 1010, New Zealand; 5Faculty of Medical and Health Sciences, University of Auckland, Auckland 1010, New Zealand; 6Centre for Brain Research, University of Auckland, Auckland 1010, New Zealand

**Keywords:** data augmentation, multi-class classifiers, classification, deep learning, long short-term memory neural networks, unidirectional LSTM, bidirectional LSTM, time-series aptasensor signal

## Abstract

Nanomaterial-based aptasensors are useful devices capable of detecting small biological species. Determining suitable signal processing methods can improve the identification and quantification of target analytes detected by the biosensor and consequently improve the biosensor’s performance. In this work, we propose a data augmentation method to overcome the insufficient amount of available original data and long short-term memory (LSTM) to automatically predict the analyte concentration from part of a signal registered by three electrochemical aptasensors, with differences in bioreceptors, analytes, and the signals’ lengths for specific concentrations. To find the optimal network, we altered the following variables: the LSTM layer structure (unidirectional LSTM (LSTM) and bidirectional LSTM (BLSTM)), optimizers (Adam, RMSPROP, SGDM), number of hidden units, and amount of augmented data. Then, the evaluation of the networks revealed that the highest original data accuracy increased from 50% to 92% by exploiting the data augmentation method. In addition, the SGDM optimizer showed a lower performance prediction than that of the ADAM and RMSPROP algorithms, and the number of hidden units was ineffective in improving the networks’ performances. Moreover, the BLSTM nets showed more accurate predictions than those of the ULSTM nets on lengthier signals. These results demonstrate that this method can automatically detect the analyte concentration from the sensor signals.

## 1. Introduction

Aptamer-based biosensors have been widely used in various fields, such as environmental monitoring [1], food quality and safety [2], and medical diagnostics and therapy [3], due to the physical and chemical features of aptamers for detection and small binding substances [4]. Moreover, the advancement in nanostructured materials has attracted much attention in recent years due to their potential applications and unique properties, including high reactivity, high functionalization, large surface-area-to-volume ratio, and small size [5]. Thus, advanced nanostructured materials have been utilized to improve the sensing capacities of aptasensors [6], lower the limits of detection of analytes [7], and amplify the sensors’ signals [8]. Nanomaterial-based aptasensors have been used as effective instruments for recognizing small analytes in clinical health diagnostics [6,7], medical therapy [9], and disease biomarker detection [10]. In addition, using and improving these analytical devices for identifying and quantifying a target analyte is beneficial due to their having higher specificity and selectivity and their elimination of labor-intensive and time-consuming procedures, expensive instruments, and multiple analytical steps [11].

Applications of machine learning (ML) algorithms have been widely used in healthcare as powerful tools for creating prediction models and making precise decisions [12]. Consequently, machine learning has significantly improved biosensors, such as through the analysis of sensing data for anomaly detection, noise reduction, classification, and pattern recognition [13]. Identifying ultra-low levels of biological species is a critical objective in improving biosensors in medical diagnostics and therapy [14].

Deep learning (DL) algorithms, as a subcategory of machine learning algorithms, have progressed remarkably on broad datasets with distinctive modalities, including time series, images, and natural languages [15]. Significant progress in deep learning has been beneficial for solving problems in many domains, including the medical and healthcare fields, and it has also defeated the conventional machine learning models [15]. For example, convolutional neural networks (CNNs) are a better fit for problems dealing with image processing, and recurrent neural networks (RNNs) are suitable for modeling problems that require the processing of time series or sequential data [16].

There are different RNN-based networks, and their main distinguishing feature is the difference in how they remember the input data [16]. For example, an original RNN is incapable of remembering past data, while a long short-term memory (LSTM) network is a modified version of RNN-based networks capable of remembering and learning from past data [16]. This means that LSTMs are suitable for making a prediction model when the datasets are in the form of time series due to their ability to learn temporal dependencies by employing a gating mechanism for data analysis [17]. Moreover, LSTM solves the vanishing gradient problem, unlike the original RNN [18].

LSTM networks have a broad range of applications in data processing and decision-making in healthcare [19,20]. Saltepe et al. [21] utilized two LSTM networks to detect and classify gold ion concentrations ranging from 0 to 100 μM to decrease the time needed for the detection of gold ions, which ranged from 30 min to 3 h. The first network was a binary classification network and was designed to detect the existence of gold ions in the sample. This network detected the gold ions’ presence in the sample with prediction accuracies of 78% and about 98% from the 30-min and 3-h time series, respectively. The second LSTM network that was designed for classifying the gold concentrations showed a prediction accuracy of 82% from the 3-h time series. Klosowski et al. [22] applied two LSTM networks to classify six types of heart dysfunctions based on ECG time-series signals. The LSTM networks used the raw and the double spectral ECG data, and the accuracies of their predictions were 70.8% and 100%, respectively.

Some research has been conducted regarding the classification of biosensors’ signals [21,23]. The similarity among these studies is that the analyte concentration remained constant from the start to the end of the signal registration. This means that signals of different analyte concentration levels were recorded separately and independently. To the best of our knowledge, there has not been enough research on the detection of concentration from electrochemical biosensor signals in conditions in which signals are registered while the concentration of the analyte increases after a specific time and in which the sensing platform does not show that it is sensing the analyte anymore. Finding a suitable workflow for classifying these biosensor signals motivates this research, since it might ease the development of biosensor. Thus, this study presents a deep learning model composed of recurrent LSTM layers that are capable of classifying analyte concentrations from aptasensor signals.

In this paper, we utilized LSTM networks to analyze the transient signals from three nanomaterial-based aptasensors in order to detect the presence and concentrations of target analytes and automate signal classification regarding the analyte concentration. In addition, all computations and deep learning algorithms were implemented in MATLAB 2021b. The following describes the workflow followed in this paper. First, a preprocessing technique was applied to the sensor signals. Then, the preprocessed original signals were split into two categories: original data (OD) network data and control group data. In the next step, the OD network was used to make prediction models, and the control group was used to assess the prediction models. Then, a data augmentation method was developed to increase the data size. Then, both original and augmented data were used to make and train the LSTM models, and the test set and control group were responsible for assessing the performance of the prediction models.

## 2. Materials and Methods

This section describes the methods applied in this study, including those for data collection, data preprocessing, data augmentation, LSTM architecture and optimization, and evaluation of the prediction models.

### 2.1. Dataset Descriptions

The datasets used in this study were (1) 35-mer adenosine, (2) 31-mer oestradiol, and (3) 35-mer oestradiol. The datasets contained several time-series signals representing the drain current of three different aptasensors. Table 1 describes three key features of the sensors used for data collection to provide a quick and brief comparison of the datasets. The distinguishing features of these datasets were two main components of their sensors: their target analytes and their bioreceptors. However, their transducers, another main component used for these aptasensors, were carbon nanotube (CNT) field-effect transistors (FETs). Explaining all of details of the functionalization of these sensors is beyond the scope of this paper. However, detailed information on the 35-mer adenosine sensor, including transistor fabrication and aptamer functionalization, can be found in [24].

As the sensing protocols for the drain current measurements might provide a clear insight into the registered signals, the following explains the method of measuring the signals. The sensing protocols for measuring the 31-mer and 35-mer oestradiol sensors’ responses were similar, but they were different from those of the adenosine sensors. Table 2 summarizes and compares the sensing protocols of the adenosine and oestradiol datasets.

The sensing responses for adenosine and oestradiol were measured in time intervals of 1 and 1.081 s with a standard deviation of $5\times 10^{-3}$, respectively, with similar gate and drain voltages, i.e., $V_{G}=0$ V and $V_{D}=100$ mV. The buffer selected for the adenosine sensor was 2 mM Tris-HCI, and that for the oestradiol sensors was 0.05 × PBS (phosphate-buffered saline) with 5% ethanol (EtOH).

Regarding the adenosine sensor, the initial load for each measurement was 110 μM of 2 mM Tris-HCI in a polydimethylsiloxane (PDMS) well, which lasted for 1000 s. Then, the adenosine solution was added to the PDMS well every 500 s in successively greater concentrations, considering the adenosine concentration in the PDMS well before each addition. The process of adding the adenosine solution increased the adenosine concentration, which varied from 1 pM to 10 μM in the PDMS well.

Regarding the oestradiol sensors, the initial load for each measurement was 100 μL of 0.05 × PBS 5% EtOH in the well, which lasted for 300 s. Then, in the next 300 s, 20 μL of 0.05 × PBS 5% EtOH was added, while the oestradiol concentration did not increase. Then, the oestradiol solution was added to the well every 300 s in successively greater concentrations, considering the oestradiol concentration in the well before each addition. In addition, each time the oestradiol concentration was increased, a solution of 20 μL of 0.05 × PBS 5% EtOH was added to the well. The process of adding the oestradiol solution increased the oestradiol concentration, which varied from 1 nM to 10 μM in the well.

### 2.2. Contextual Outlier Detection

A contextual outlier, also known as a conditional anomaly, is defined as a data instance whose pattern does not conform to that of other well-defined data instances with similar contextual information [27,28]. Regarding the experiments related to this study, factors that could cause contextual outliers were background noises in the lab, the use of broken transistors, issues in immobilizing the aptamers on carbon nanotube surfaces, issues in fabricating the sensing interface, and so on.

The patterns of signals affected by these factors deviate from the patterns of well-defined and normal signals. The purpose of removing outliers is to eliminate non-informative signals or segments. As there were a few signals in the datasets, removing the outliers was performed with prior knowledge of the biosensors’ behaviors and through data visualization.

In this paper, the signals were preprocessed with data normalization before being fed into the DL models. It needs to be mentioned that data normalization was applied to the entire signal. Data normalization or feature scaling puts all of the signals in a dataset on the same scale and prevents a feature from dominating and controlling the others. The data normalization applied in this paper was a Z-score scaling that used the mean (μ) and standard deviation (σ) of a signal.

Suppose that $X=[x_{1},x_{2},\ldots ,x_{i},\ldots ,x_{n}]$ is an entire signal, where the *n* is the number of data points within the given signal or the length of the signal. Then, Equation (Equation 1) shows the new signal $\hat{X}$ created by Z-score scaling.
(1)$$\hat{X}=\left[{\hat{x}}_{1},{\hat{x}}_{2},\ldots ,{\hat{x}}_{i}=\frac{x_{i}-\mu }{\sigma },\ldots ,{\hat{x}}_{n}\right].$$

#### Segmentation and Labeling

After rescaling the signal, it was split into different segments. Each segment was a part of the signal for which the concentration of analyte remained constant from its beginning to its end. Then, each segment was labeled with its corresponding analyte concentration. This means that the labels for the three datasets were: No Analyte, 1 nM, 10 nM, 100 nM, 1 μM, and 10 μM. As shown in Table 3, these six labels and concentrations were considered as the six different classes, in the same order.

### 2.3. Data Split

The data fed into the DL model needed to be split into three subsets, namely, the training, validation, and test set, with the proportions of 60%, 20%, and 20%, respectively. The training set was used to extract meaningful information and find the optimal parameters, the validation set was used to tune the parameters, and the test set was used to assess the model’s performance [13].

In this paper, the original data were split into two sets—network and test sets—with proportions of approximately 70% and 30%, respectively. These sets were named the original data (OD) network and OD test set, respectively. The former set was used to make DL models based on original data and for data augmentation. The latter, the OD test set, assessed the DL models and acted as a control group. The reasons for considering this OD test set were to prove the functionality of the data augmentation method in making the prediction models and to avoid biased results. In order to complete the information related to the data split, it must be mentioned that the augmented data and OD networks were randomly shuffled and separated again into the training set (60%), validation set (20%), and network test set (20%).

### 2.4. Data Augmentation

In machine learning, small amounts of training data might cause overfitting and might not be enough for training models [29]. The need for data augmentation is more critical for real-world data, since acquiring large enough real-world datasets has not always been possible due to cost or time limitations. Generating synthetic data, which is also known as data augmentation, is a solution for overcoming the problem of insufficient data samples [30] or compensating for datasets with imbalanced classes [31]. Data augmentation helps to increase the generalization capability of an ML prediction model and improve the model’s performance by increasing the variability of the training and validation data samples [32,33].

In this paper, we utilized a data augmentation method to increase the size of the available datasets. Suppose the ${\hat{S}}_{i}$ and ${\hat{S}}_{j}$ are two preprocessed segments from an identical dataset with similar analyte concentrations. Then, $S_{aug}$ is an augmented segment generated with the following Equation (Equation 2): (2)$$S_{aug}=w\times {\hat{S}}_{i}+\left(1\;-\;w\right)\times {\hat{S}}_{j}\;,$$
where w∈(0,1) and is a normally distributed random number generated by the randn function in MATLAB R2021b.

### 2.5. Background of LSTM

This subsection explains long short-term memory (LSTM) and its architecture. Then, the unidirectional and bidirectional LSTM structures, as well as their similarities and differences, are discussed.

The advantage of using an LSTM network over a recurrent neural network (RNN) is that LSTM can capture the temporal dependency of input sequences during the training process [21,34]. An LSTM network is an RNN that prevents the long-term dependency problem by utilizing gates and calculating a hidden state with an enhanced function [17]. The building blocks of LSTM networks are LSTM cells, which means that an LSTM layer consists of recurrently connected cells [16,17,34]. An LSTM cell, or an LSTM hidden unit, consists of four parts: the forget gate, input gate, output gate, and a cell candidate. Figure 1 presents the structure of a cell. This cell decides to ignore or remember something in its memory by using a gating mechanism. The role of the three gates is to selectively control and transfer needed information into and out of the cell [34]. This figure can also be considered an LSTM layer consisting of only one memory cell or hidden unit, where $X_{t}$ and $h_{t}$, respectively, are the input and output of the LSTM layer.

Consider $X_{t}$ as a sequence input into the memory block at time step *t*; the forget gate selects which data to erase and which data to remember. As shown in Equation (Equation 3), these decisions are made by the sigmoid layer:(3)$$f_{t}=\sigma \left(W_{fx}\;X_{t}+W_{fh}\;h_{t-1}+b_{f}\right).$$

The input gate is responsible for controlling the level at which the cell state is updated by using another sigmoid layer. As shown in Equation (Equation 4), the sigmoid layer of the input gate decides which data need to be updated. In the next step, as shown in Equation (Equation 5), the cell candidate (${\tilde{c}}_{t}$) is responsible for adding information to the cell state by using the tanh layer. Now, the cell state is ready to be updated with the combination of the forget and input gates and new candidate values of ${\tilde{c}}_{t}$. Equation (Equation 6) describes the mathematical formula for calculating the cell state: (4)$$i_{t}=\sigma \left(W_{ix}\;X_{t}+W_{ih}\;h_{t-1}+b_{i}\right),$$
(5)$${\tilde{c}}_{t}=tanh\left(W_{cx}\;X_{t}+W_{ch}\;h_{t-1}+b_{c}\right),$$
(6)$$c_{t}=f_{t}\times c_{t-1}+i_{t}\times {\tilde{c}}_{t}.$$

The output gate, which is shown in Equation (Equation 7), utilizes a sigmoid layer to decide which part of the cell state contributes to the output. Now, the hidden state or output of the memory cell is ready to be calculated. The output gate and the cell state are contributors to the hidden state. Equation (Equation 8) presents its mathematical formula: (7)$$o_{t}=\sigma \left(W_{ox}\;X_{t}+W_{oh}\;h_{t-1}+b_{o}\right),$$
(8)$$h_{t}=o_{t}\times tanh\left(c_{t}\right).$$

Note that $W_{ix}$, $W_{fx}$, $W_{ox}$, and $W_{cx}$ refer to the input weight matrices for the input gate, forget gate, output gate, and cell value, respectively, and $W_{ih}$, $W_{fh}$, $W_{oh}$, and $W_{ch}$ are the recurrent weights for the gates and the cell value in the same order. Their corresponding bias vectors are $b_{i}$, $b_{f}$, $b_{o}$, and $b_{c}$.

Moreover, it can be seen that the cell state and gate activation functions (AFs) are, respectively, tanh (Equation (Equation 9)) and sigmoid (Equation (Equation 10)); these map the nonlinearity and make decisions: (9)$$tanh\left(z\right)=\frac{e^{2z}-1}{e^{2z}+1},$$
(10)$$\sigma \left(z\right)=\frac{1}{1+e^{-z}}.$$

An LSTM layer in a deep neural network consists of a set of LSTM cells. LSTM layers can be categorized into unidirectional LSTM (ULSTM) and bidirectional LSTM (BLSTM) layers. Figure 2 represents a ULSTM structure. It can be said that a ULSTM structure is an RNN that uses LSTM cells instead. The unfolded figure of the ULSTM shows that the output of each cell is the input for the next cell in the same layer. It should be mentioned that an LSTM block refers to several LSTM cells or hidden units.

Figure 3 depicts a BLSTM structure consisting of forward and backward layers. The unfolded figure shows that the forward layer moves in a positive temporal direction, while the backward movement is in a negative temporal direction. In addition, the outputs from both the forward and backward LSTM cells are joined and concatenated as the layer’s output.

Figure 4 presents the flow of information in an LSTM layer during different time steps. In this figure, *N* is the length of the sequential input for the LSTM layer, *L* is the number of hidden units in the LSTM layer, and *T* is the length of the training set. Note that in this figure, $h_{t}$ can be considered as just the forward movement ($\vec{h_{t}}$) in the ULSTM layer or as a concatenation of both forward ($\vec{h_{t}}$) and backward ($\overset{\leftarrow }{h_{t}}$) movements in the BLSTM layer.

### 2.6. LSTM Architecture

In this work, two LSTM networks were employed to classify the analyte concentrations. The objective was to classify the input data into six different concentration classes: 0 M, 1 nM, 10 nM, 100 nM, 1 μM , and 10 μM. The target outputs of each class were labeled in a binary vector format, where the desired class was labeled with “1” and the others were labeled with “0”. Recall that the input data were the corresponding concentration segments of the signals, as well as the original and/or augmented segments.

Figure 5 visualizes the architectures of both networks. The networks comprised five successive layers: a sequential input layer, an LSTM layer, a fully connected layer, a softmax layer, and a classification layer. The only difference between the networks was in their LSTM layers. The LSTM layer in the first network was a unidirectional LSTM layer, while this was a bidirectional LSTM layer in the second network. It should be taken into consideration that the fully connected (FC) layers were affected by the previous LSTM layers and the number of output classes in the classification layer.

Table 4 describes and compares the layers and the properties of the unidirectional and bidirectional LSTM networks depicted in Figure 5. Recall that the size of the input layer entering the networks was equal to the length of the segments and was considered as one sequence, and the output size of the networks (*m*) was identical to the number of classes in the data. It should be mentioned that all of the input weights, the recurrent weight, and the bias matrices were concatenated together to form the input weights ($W_{x}=\left[W_{ix};W_{fx};W_{cx};W_{ox}\right]$), recurrent weights ($W_{h}=\left[W_{ih};W_{fh};W_{ch};W_{oh}\right]$), and bias ($b=[b_{i};b_{f};b_{c};b_{o}]$).

### 2.7. LSTM Optimization

We adopted the following variables to find the optimal prediction model: the number of hidden units in the LSTM layers, the optimizers (Adam, RMSPROP, and SGDM) [35], and the number of segments fed into the networks. Initially, the original datasets were used to train the networks with the total size of a segment. The altered variables were: the number of hidden units in the LSTM layers—starting from 50 and increasing to 500 in increments of 50—and the use of the three optimizers mentioned. In the next step, we used both the original and augmented signals to train the networks. The sums of the original and augmented segments per class were 50 and 100.

It should be noted that all deep learning algorithms were implemented with the MATLAB R2021b Deep Learning Toolbox.

### 2.8. Evaluation Metrics

After training the prediction models, the classification performance of the neural networks needed to be assessed with relevant metrics. In this paper, the LSTM networks’ performances were assessed using two standard metrics: overall accuracy (ACC) and macro F1-score (MF1) [21,36]. In this work, we created a confusion matrix using the predictions from the test data, and then the overall accuracy and the macro F1-score were calculated with the confusion matrix.

The overall accuracy was calculated from the sum of the diagonal numbers of the confusion matrix divided by the total number of elements. In other words, the overall accuracy (Equation (Equation 11)) was the proportion of correctly classified elements among all elements in the test set.
(11)$$\mathrm{ACC}=\frac{\mathrm{Number}\;\mathrm{of}\;\mathrm{true}\;\mathrm{testing}\;\mathrm{outputs}}{\mathrm{Total}\;\mathrm{number}\;\mathrm{of}\;\mathrm{elements}\;\mathrm{in}\;\mathrm{the}\;\mathrm{test}\;\mathrm{set}}.$$

Before mentioning the method for calculating the macro F1-score, its building blocks—recall, precision, and F1-score—need to be defined. The recall, which is the true positive rate (TPR) and is presented in Equation (Equation 12), is the number of correctly classified positive elements among all positive elements. Precision, which is the positive predicted value (PPV) and is presented in Equation (Equation 13), is the number of correctly classified positive elements among the elements classified as positives by the model. Then, the harmonic mean of the recall and precision is called the F1-score (Equation (Equation 14)). The macro F1-score (Equation (Equation 15)) is the mean of the class-wise F1-score of each concentration.
(12)$$\mathrm{Recall}=\mathrm{TPR}=\frac{\mathrm{TP}\;}{\mathrm{TP}\;+\;\mathrm{FN}},$$
(13)$$\mathrm{Precision}=\mathrm{PPV}=\frac{\mathrm{TN}}{\mathrm{TP}\;+\;\mathrm{FP}},$$
(14)$$F1\text{-}\mathrm{score}=\frac{2}{1/\mathrm{precision}+1/\mathrm{recall}},$$
(15)$$\mathrm{Macro}\;F1\text{-}\mathrm{score}=\mathrm{MF}1=\frac{1}{m}\sum_{i=1}^{m}{\left\{F1\text{-}\mathrm{score}\right\}}_{i},$$
where *m* is the number of classes in a given dataset.

## 3. Results

### 3.1. Datasets

Figure 6 presents the typical raw signals of the available datasets. Vertical dashed lines separate the different analyte concentrations (ACs). Note that the initial ACs for the 35-mer adenosine experiments were not necessarily identical. Figure 6A,B represent the drain current in two experiments from the adenosine dataset in which the initial AC for the former experiment was 1 nM and that for the latter one was 1 μM. However, the initial ACs for all of the experiments of the 31-mer and 35-mer oestradiol datasets were completely similar; these are depicted in Figure 6C,D, respectively.

In addition, we need to clarify the notions of an entire signal and a segment, as these notions will be repeatedly used in the rest of this study. An entire signal refers to all of the data points registered from the beginning to the end of an experimental measurement. For example, in Figure 6A, the entire signal comprises the data points at t∈[1,3500]. On the other hand, a segment refers to a part of a signal that represents the sensor response regarding a specific analyte concentration. For example, Figure 6B contains three segments: the No Analyte segment for t∈[1,1000], the 1μM segment for t∈[1001,1500], and the 10μM segment for t∈[1501,2000].

### 3.2. Contextual Outlier Detection

Figure 7 compares the normal signals with the contextual outliers. Figure 7A shows the normal pattern of the 35-mer oestradiol signals that were registered with a well-fabricated device. It can be seen that these signals had similar trends when the analyte concentration increased. On the other hand, Figure 7B shows that the signals that did not conform to normal signals. The red and blue lines were registered with non-sensing and broken transistor devices. All of the signals that were registered by the non-sensing and broken transistor devices were completely removed from the datasets.

In addition, Figure 7C shows two instances of signals that showed temporal abnormalities although the sensors were normal and other parts of the signals showed reasonable sensing. In this figure, the drain currents registered between 300–750 s were considered the contextual outliers. Thus, there was partial removal of the signals that were related to the abnormal segments. Consequently, the other segments of the signals remained in their relevant datasets for utilization in the deep learning models.

**Figure 7 bioengineering-09-00529-f007:**
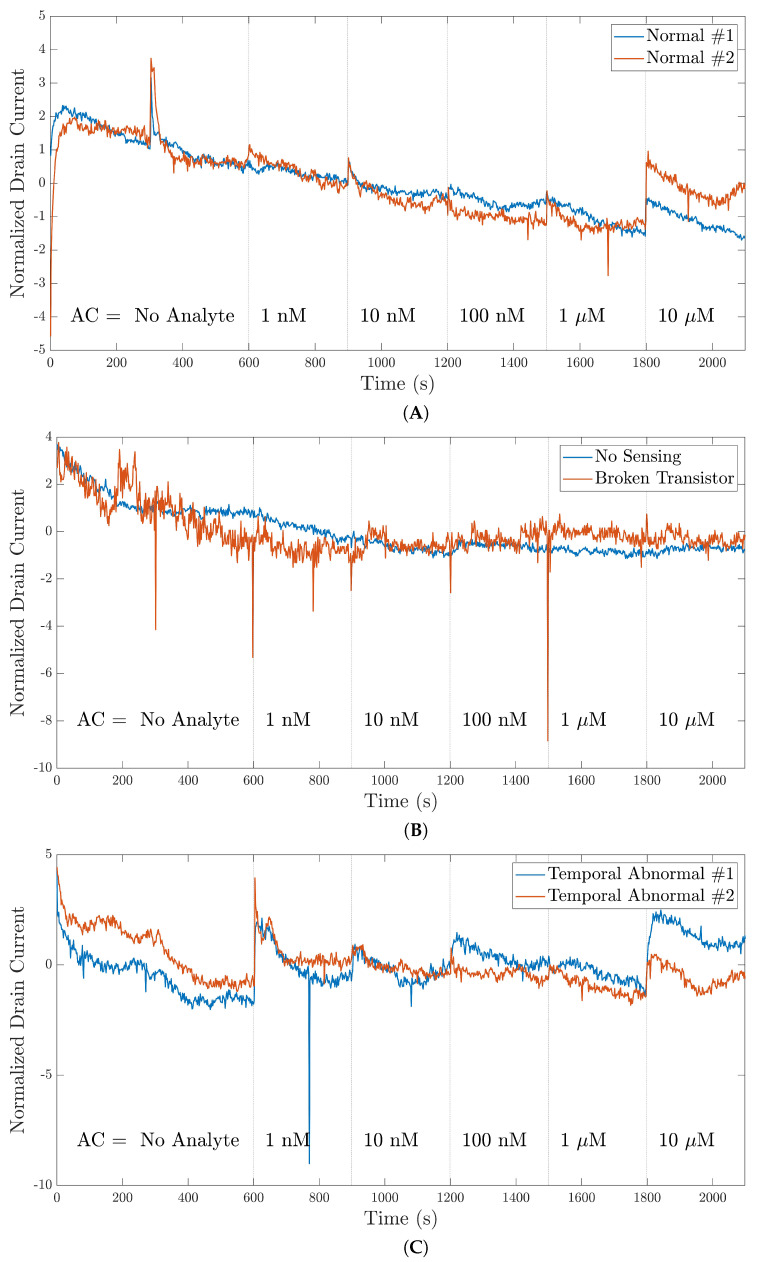
The plots compare the normal behavior in the 35-mer oestradiol signals with the behavior of the contextual outliers: (**A**) signals with normal patterns; (**B**) signals with contextual outliers (the red and blue lines in the plot represent signals generated by non-sensing and broken transistor devices, respectively); (**C**) abnormal temporal signals that were registered with normal sensors but showed unusual behavior between 300 and 750 s.

It should be noted that removing the segments that showed temporal abnormalities was performed after data normalization, since the mean and standard deviation used for data normalization depended on the information of an entire signal.

Table 5 shows the number of available normal segments in each dataset that were considered normal and suitable for utilization in the DL networks. However, as there were insufficient segments of adenosine with a concentration of 100 pM, these segments were excluded from the adenosine dataset.

### 3.3. Data Preprocessing

Figure 8 presents and compares the instances of preprocessed signals according to the Z-score scaling. Recall that this preprocessing method was applied to the entire signal. Figure 8A shows a raw signal from the 35-mer adenosine dataset, and Figure 8B shows the preprocessed signal according to Equation (Equation 1).

### 3.4. Data Split

Table 6 presents the numbers of segments used in the OD network and the OD test sets in each dataset after the original data split. The network sets were used for feeding into the networks, and the OD test sets, which consisted of original data, were used to assess the networks. The data split in the oestradiol datasets was done by considering three and four entire signals for the 31-mer and 35-mer as their test sets, respectively. However, the data split in the 35-mer adenosine dataset was not as straightforward as that for the oestradiol datasets. The OD test set was selected segment-wise.

### 3.5. Data Augmentation

Figure 9 presents the augmented segments generated from the normalized drain currents.

### 3.6. LSTM Optimization

Table 7 fully describes the model hyperparameters for optimizing the ULSTM and BLSTM. Moreover, the MATLAB Deep Learning Toolbox set the other hyperparameters and functions that are not mentioned in this table to their default values. For example, the function used to initialize the input weights was the default function, i.e., the glorot weight initialization function.

### 3.7. Evaluation

It was mentioned that we used the overall accuracy (ACC) and macro F1-score (MF1) as the performance metrics. Moreover, all of the networks were assessed with two datasets: the network test set and the OD test dataset. This means that there were two tables of evaluation results for each dataset after altering the effective parameters for optimization. Thus, these tables were organized according to the network and OD test datasets.

Table 8 and Table 9 present the performance metrics of the test and control data, respectively, from the networks trained with the 35-mer adenosine dataset. Table 10 and Table 11 present the performance metrics for the network and control sets from the networks trained with the 31-mer oestradiol dataset, respectively. Table 12 and Table 13 present the evaluation metrics for the test data when trained with the 35-mer oestradiol signals.

For a better understanding of the tables, the boldface numbers indicate the highest performance of a network based on the optimizers and the LSTM layer structure for 50 and 100 augmented segments. Moreover, we ignored the best classifiers when the numerical metrics for almost all hidden units were similar. For example, in Table 9, the best classifiers were not selected for accuracy when using 50 segments in the ULSTM and BLSTM structures and the SGDM optimizer. The reason for this was that the use of more than four hidden units resulted in the same accuracy.

Figure 10 shows the overall accuracy of the OD test set from the network trained with the original segments from the 35-mer adenosine dataset. The accuracies of the prediction models trained with the original data varied approximately from 30% to 55% for the three datasets.

Figure 11 depicts the effect of increasing the number of augmented segments on the prediction models. This figure shows the accuracy of the test data from the networks trained with the ADAM optimizer and the 35-mer adenosine dataset. In general, it can be seen that the performance of the prediction models significantly improved by utilizing the data augmentation method. However, the prediction models were slightly improved with the increase in the amount of augmented data.

Figure 12 shows the overall accuracy from the control data of the ULSTM and BLSTM networks trained with the 35-mer adenosine dataset when 100 augmented segments per class were used.

In more detail, regarding the 35-mer adenosine dataset, as shown in Table 8 and Table 9, the MF1 for the original data changed approximately from 50% to 72%. Then, the ACC and MF1 reached 85% and 94%, respectively, by utilizing the augmented segments for the control data, as shown in Table 9. For the 31-mer oestradiol dataset, Table 10 and Table 11 show that the accuracy reached 90% and the MF1 increased approximately from 50–83% to 54–93%. For the 35-mer oestradiol dataset, Table 12 and Table 13 depict that the accuracy range of 25–50% reached 70% by using the augmented data, and the MF1 increased from the range of 30–63% to 76% after data augmentation.

## 4. Discussion

Nanomaterial-based aptasensors are useful biosensors that are capable of detecting small chemicals and species. A vital goal in the advancement of biosensors is the identification and measurement of low levels of target analytes. Deep learning methods are attractive tools for the advancement of biosensor technology and the analysis biosensing data. RNNs exploit temporal information in time-series inputs to make prediction models for classification and regression problems.

In our work, we successfully employed LSTM networks to automatically predict analyte concentrations from parts of drain current signals registered by three different electrochemical aptasensors. The differences in these sensors were their bioreceptors, their analytes, and the lengths of the signals for specific concentrations. Among RNN-based models, LSTM networks, which contain a gating schema for data analysis, are suitable models due to their ability to learn temporal dependencies. Thus, we utilized ULSTM and BLSTM networks with different optimizers and hidden units to identify the optimal classification models for various concentrations.

Moreover, similarly to most real-world problems, the available signals registered by these sensors were insufficient for training the networks. To overcome this limitation, we proposed a data augmentation method in order to increase the size of the available datasets and improve the prediction model’s generalization ability and overall performance. The augmentation method improved the model’s performance.

By comparing the evaluation results obtained with the original and augmented data, it can be seen that the applied data augmentation method significantly improved the classification performance. In addition, the results showed that the number of hidden units might not be very effective in enhancing LSTM models. Considering the optimizers, the LSTM networks that used the SGDM optimization algorithm showed a lower prediction performance than that obtained with the ADAM and RMSPROP algorithms [35].

According to the results showing the networks’ performance on the test data, the BLSTM networks used for the three datasets presented more accurate classification than that of the ULSTM networks [37]. However, this result was not identical to that obtained with the OD test data on the 31-mer and 35-mer oestradiol datasets, i.e., the prediction performance of the BLSTM models on both the test and control segments of the 35-mer adenosine dataset was higher than that of the ULSTM structures, but not for the oestradiol datasets. The reason for this failure to improve the performance of the BLSTM over that of the ULSTM for the oestradiol datasets might be the shorter length of the oestradiol segments compared to that of the adenosine segments.

Regarding the datasets, we observed that the least accurate models belonged to the 35-mer oestradiol dataset, and the most accurate networks were trained with the 35-mer adenosine dataset. This low accuracy might have occurred for one of two reasons: the aptamer length used to detect the analyte or the criteria chosen for the detection of contextual outliers. It was shown that successful detection of the analyte depended on the aptamer length, and the analyte–aptamer binding must occur within the Debye length to affect a CNT FET’s drain current [38]. Thus, the 35-mer oestradiol signals might not have shown obvious evidence of sensing. In the case that this low-accuracy model might be the result of the detection of contextual outliers, it can be assessed in a future study.

In future work, we can evaluate the effects of the segment length and automatic detection of contextual outliers on the prediction models. In order to address the unequal size of segments in future studies, data augmentation methods based on artificial neural networks might be a solution. In fact, by defining a regression model and employing deep learning networks [39], we can extend the length of available time-series data. Consequently, this technique can produce new datasets containing segments of similar lengths. Regarding the detection of contextual outliers, neural networks such as autoencoder-based [40] or LSTM-based models [41] can be utilized to assess and compare the effects of automatic anomaly detection on the prediction models.

The insufficiency of the data exposed a limitation in the preprocessing method applied in this work. Typical data preprocessing methods apply identical changes to the training and test sets. However, we used Z-score scaling for each signal based on its mean and standard deviation. It was impossible to estimate the mean and standard deviation with the available statistical methods, such as with simulations and hypothesis tests.

## 5. Conclusions

In this work, we exploited a data augmentation method and LSTM networks to analyze the drain current signals of three similar CNT FET aptasensors in the 35-mer adenosine dataset and 31-mer and 35-mer oestradiol datasets. The drain current signals reflected the sensing responses of the aptasensors, while the concentrations of target analytes successively increased from 1 nM to 10 μM. The ultimate purpose of analyzing the signals was to automatically detect and classify the analyte concentration according to the corresponding sensing response.

The scaling-based data augmentation method was proven to be highly effective for improving and increasing the generalization ability of LSTM-based classification models. In addition, the results suggest that the applied data augmentation method might be more effective and useful in capturing the features from lengthier signals, e.g., 35-mer adenosine signals.

Moreover, the LSTM networks were successful in sensing response classification and in predicting the analyte concentration. In addition, the results suggest that using BLSTM networks does not necessarily result in making more accurate prediction models when using augmented data. Higher scores for evaluation metrics, namely, accuracy and MF1, of the BLSTM over the ULSTM might have resulted due to overfitting. Thus, using a control group when evaluating prediction models seems to be vital in order to obtain robust results. In addition, changing the number of hidden units was not effective with respect to the performance of the prediction models, and the SGDM algorithm was not a suitable optimisation algorithm for them.

## Figures and Tables

**Figure 1 bioengineering-09-00529-f001:**
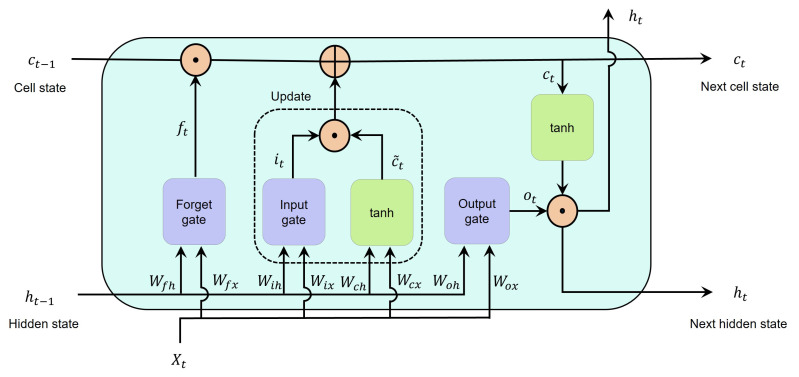
Illustration of an LSTM cell. Note that ⊙ and ⊕ refer to element-wise multiplication and addition, respectively.

**Figure 2 bioengineering-09-00529-f002:**
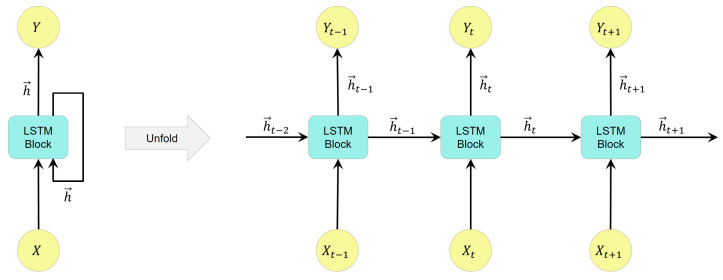
Flow of information in a unidirectional LSTM layer during different time steps with forward states, where *X*, *Y*, and $\vec{h}$ refer to the input, output, and forward states in the LSTM layer, respectively.

**Figure 3 bioengineering-09-00529-f003:**
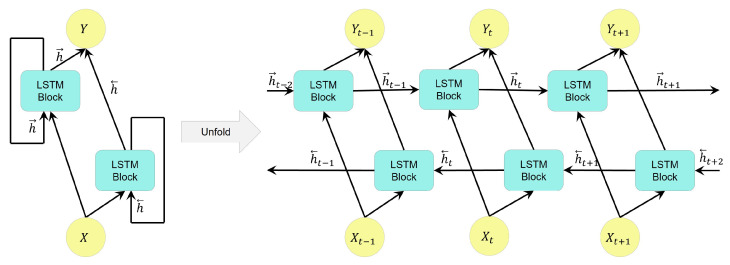
Flow of information in a bidirectional LSTM layer during different time steps with forward and backward states, where *X*, *Y*, $\vec{h}$, and $\overset{\leftarrow }{h}$ refer to the input, output, forward state, and backward state in the LSTM layer, respectively.

**Figure 4 bioengineering-09-00529-f004:**
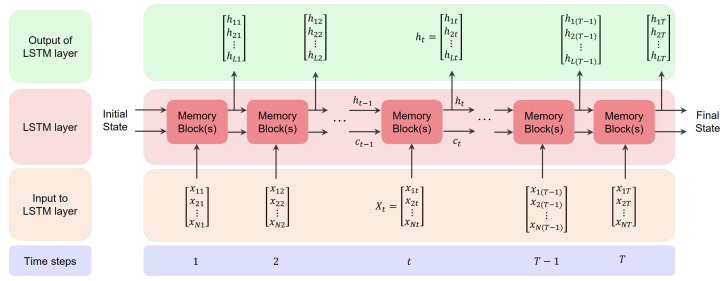
An unfolded LSTM layer that presents the flow of information during different time steps.

**Figure 5 bioengineering-09-00529-f005:**
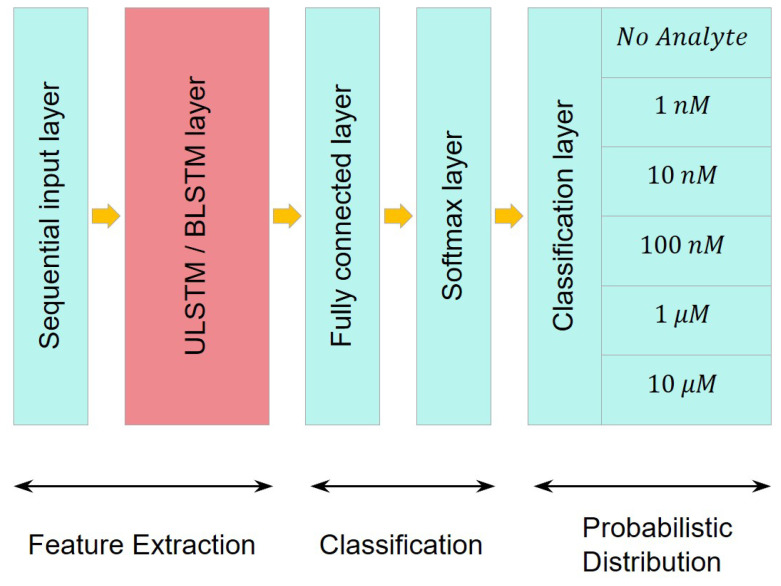
A schematic of the network architecture. Note that the only difference between the two networks was in their LSTM layer, which used either a unidirectional or a bidirectional LSTM layer.

**Figure 6 bioengineering-09-00529-f006:**
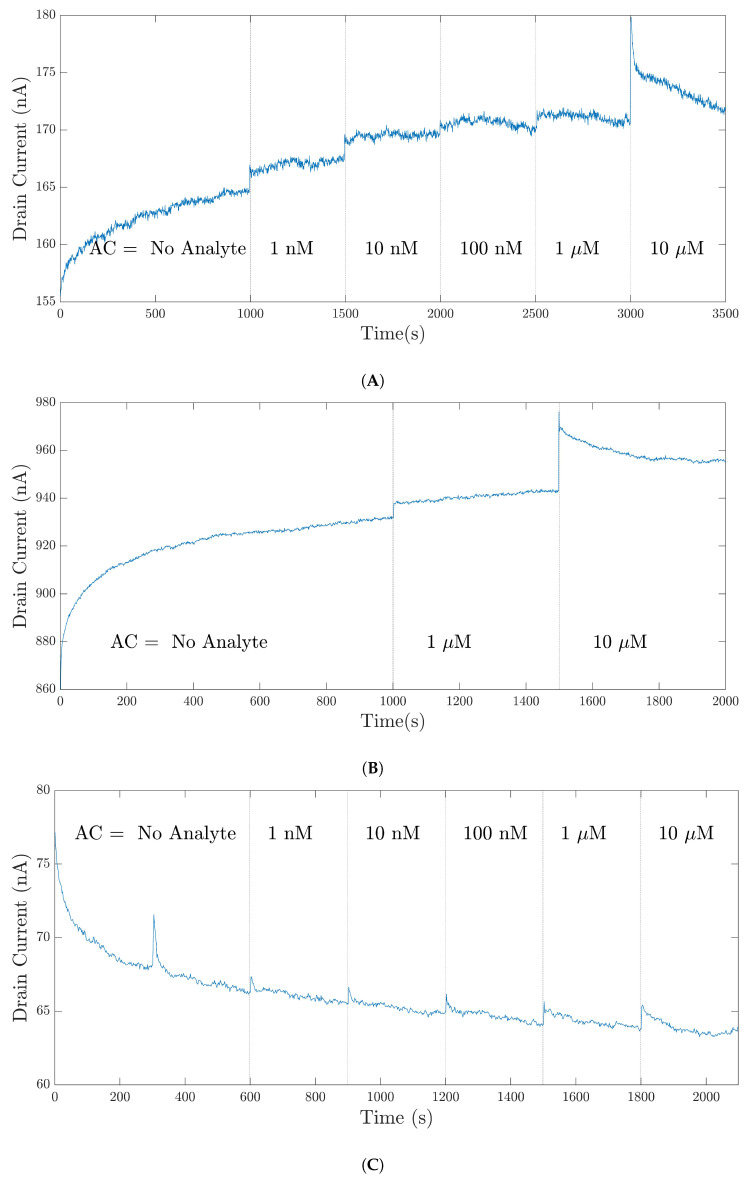
The typical raw signals of the datasets, (**A**) The 35-mer adenosine dataset with an initial concentration of 1 nM, (**B**) the 35-mer adenosine signal with an initial concentration of 1 μM, (**C**) the 31-mer oestradiol dataset, (**D**) the 35-mer oestradiol dataset. Note that in these plots, AC refers to the analyte concentration.

**Figure 8 bioengineering-09-00529-f008:**
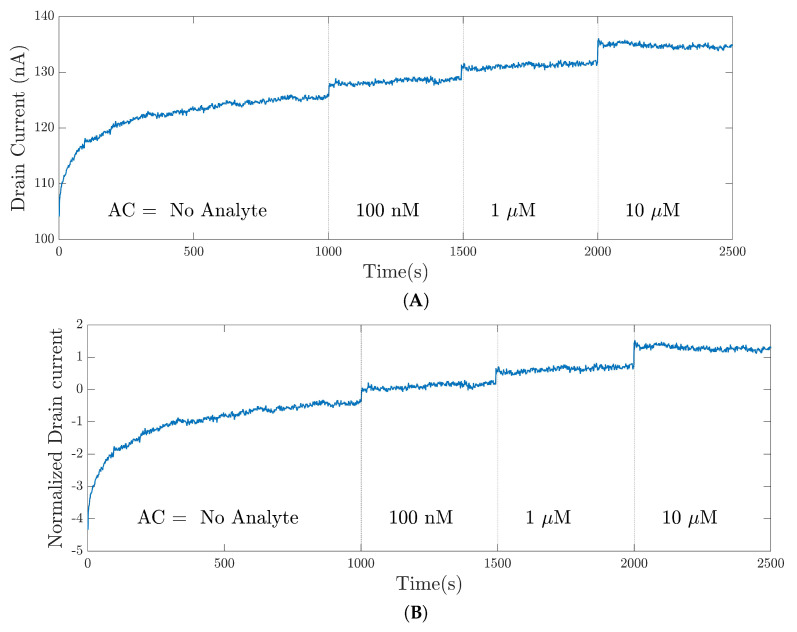
Preprocessed signal for the 35-mer adenosine dataset according to the Z-score scaling: (**A**) the raw signal; (**B**) the preprocessed signal.

**Figure 9 bioengineering-09-00529-f009:**
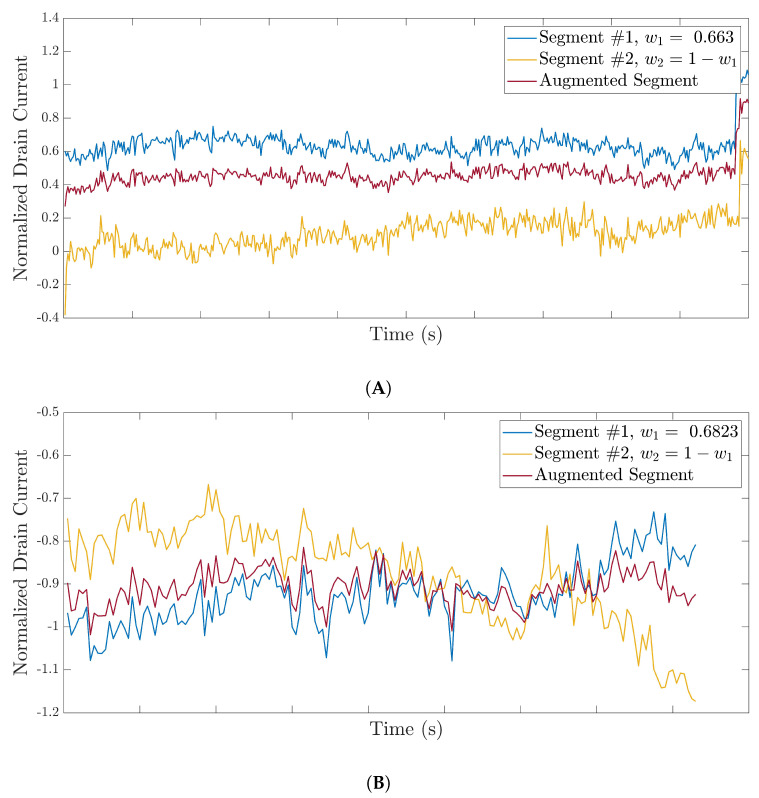
Samples of augmented normalized segments from the available datasets: (**A**) augmented segments from the 35-mer adenosine segments with a concentration of 100 nM, (**B**) augmented segments from the 31-mer oestradiol segments with a concentration of 1 μM, and (**C**) augmented segments from the 35-mer oestradiol segments with a concentration of 10 μM.

**Figure 10 bioengineering-09-00529-f010:**
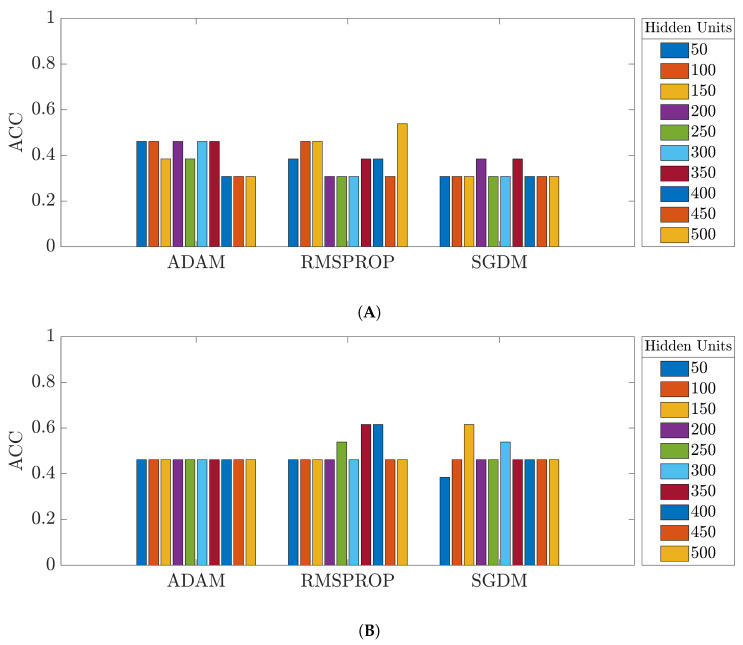
Accuracy of the networks trained with the original 35-mer adenosine dataset: (**A**) ULSTM network; (**B**) BLSTM network.

**Figure 11 bioengineering-09-00529-f011:**
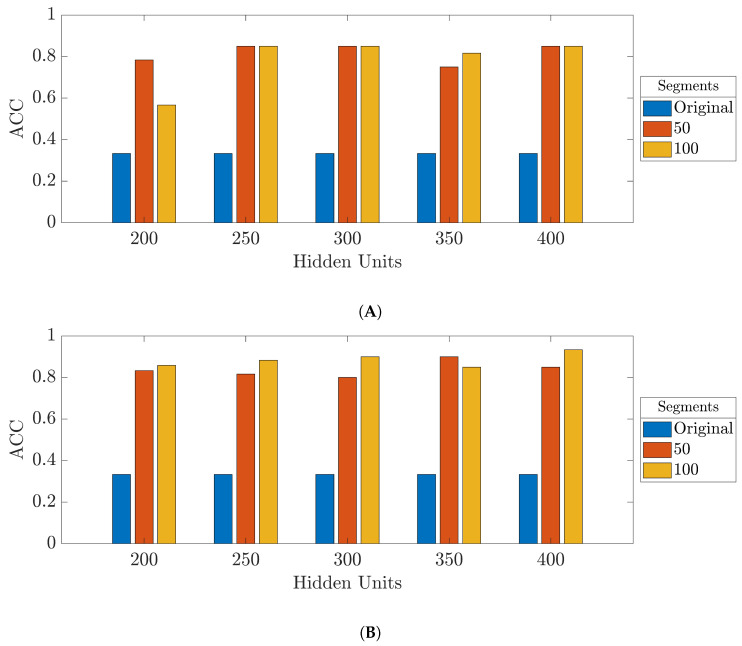
Accuracy of the test data from the networks trained with the 35-mer adenosine dataset and the ADAM optimizer: (**A**) ULSTM network; (**B**) BLSTM network.

**Figure 12 bioengineering-09-00529-f012:**
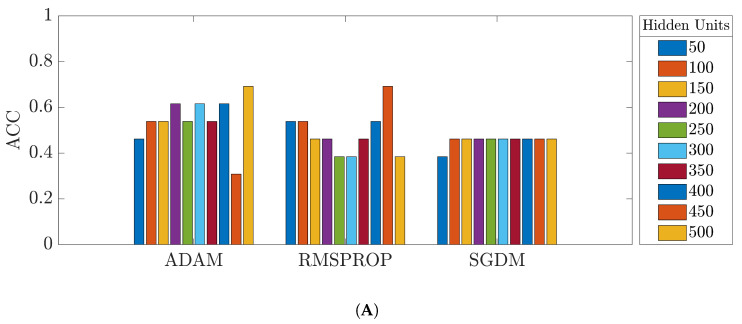
Accuracy of the control data from the networks trained with the 35-mer adenosine dataset with 100 augmented and original data points per class: (**A**) ULSTM network; (**B**) BLSTM network.

**Table 1 bioengineering-09-00529-t001:** The datasets and the main components of the sensors that recorded the signals in each dataset.

Dataset Name	Analyte	Transducer	Bioreceptor	Ref
35-mer Adenosine	Adenosine	CNT FET	$5^{\prime }$-NH2-AAAAAAAAAACCTGGGGGAGTATTGCGGAGGAAGG-$3^{\prime }$	[24]
31-mer Oestradiol	Oestradiol	CNT FET	$5^{\prime }$-GGTCCTGACCGGAGGCTGACCGGAGTGGGAA-$3^{\prime }$	[25] ^1^
35-mer Oestradiol	Oestradiol	CNT FET	$5^{\prime }$-AAGGGATGCCGTTTGGGCCCAAGTTCGGCATAGTG-$3^{\prime }$	[26]

^1^ The bioreceptor of this dataset was created by Erica S. Cassie and was a modification of that in the mentioned reference. This sequence took the common part of the best three oestradiol aptamers, and then some extra mers were added on either side.

**Table 2 bioengineering-09-00529-t002:** Comparison of the sensing protocols of the adenosine and oestradiol aptasensors. Note that the sensing protocols for both the 31-mer and 35-mer oestradiol aptasensors were completely the same. Thus, their relevant information is merged into one column.

Characteristics	Adenosine Aptasensor	Oestradil Aptasensors
Time interval of easurement	1 s	1.081 s with std $5\times 10^{-3}$
Gate voltage ($V_{G}$)	0 V	0 V
Drain voltage ($V_{D}$)	100 mV	100 mV
Buffer	2 mM Tris-HCI	0.05 × PBS with 5% EtOH
Initial step load chemical	110 μM of 2 mM Tris-HCI	100 μL of 0.05 × PBS 5% EtOH
Next steps load chemical	-	20 μL of 0.05 × PBS 5% EtOH
Initial load time	1000 s	300 s
Time interval of adding analyte	500 s	300 s
Time interval of adding chemical	-	300 s
Analyte concentration range	1 pM–10 μM	1 nM–10 μM

**Table 3 bioengineering-09-00529-t003:** The available segments’ labels and their corresponding classes.

Label	No Analyte	1 nM	10 nM	100 nM	1 μM	10 μM
**Class**	class 1	class 2	class 3	class 4	class 5	class 6

**Table 4 bioengineering-09-00529-t004:** Layer description of the LSTM deep learning model.

Layer ID	Layer Type	Hyperparameters	Learnable Parameters	State Parameters
1	Sequential input	Output size: 1	-	-
2	ULSTM	Input size: 1		
		Hidden units: *n*	$W_{x}:\;4n\times 1$	Hidden state: n×1
		Output size: *n*	$W_{h}:\;4n\times n$	
		State AF: tanh	b:4n×1	Cell state: n×1
		Gate AF: sigmoid		
	BLSTM	Input size: 1		
		Hidden units: *n*	$W_{x}:\;8n\times 1$	Hidden state: 2n×1
		Output size: 2n	$W_{h}:\;8n\times n$	
		State AF : tanh	b:8n×1	Cell state: 2n×1
		Gate AF: sigmoid		
3	FC (ULSTM) ^1^	Input size: *n*	Weights: m×n	
		Output size: *m*	Bias: m×1	-
	FC (BLSTM) ^2^	Input size: 2n	Weights: m×2n	
		Output size: *m*	Bias: m×1	-
4	Softmax	-	-	-
5	Output classification	-	-	-

^1^ Fully connected layer that follows a ULSTM layer. ^2^ Fully connected layer that follows a BLSTM layer.

**Table 5 bioengineering-09-00529-t005:** The number of available segments for each dataset after removing the contextual outlier signals and segments.

Analyte Concentration	35-mer Adenosine	31-mer Oestradil	35-mer Oestradil
No Analyte	9	9	6
100 pM	1	-	-
1 nM	4	9	6
10 nM	5	9	12
100 nM	7	9	12
1 μM	9	9	12
10 μM	9	9	12

**Table 6 bioengineering-09-00529-t006:** The number of available segments for the OD network and the OD test sets that were relevant to each dataset.

Analyte Concentration	35-mer Adenosine		31-mer Oestradil		35-mer Oestradil	
	OD Network	OD Test	OD Network	OD Test	OD Network	OD Test
No Analyte	6	3	6	3	4	2
1 nM	3	1	6	3	4	2
10 nM	4	1	6	3	8	4
100 nM	5	2	6	3	8	4
1 μM	6	3	6	3	8	4
10 μM	6	3	6	3	8	4

**Table 7 bioengineering-09-00529-t007:** The model hyperparameters for both the ULSTM and BLSTM neural networks with three optimizers.

Optimiser	ADAM	RMSPROP	SGDM
Gradient decay factor	0.9	-	-
Squared gradient decay factor	0.9	0.9	-
Momentum	-	-	0.9
Initial learning rate	0.005	0.005	0.005
Learning rate schedule	piecewise	piecewise	piecewise
Learning rate drop factor	0.2	0.2	0.2
Learning rate drop period	5	5	5
$L_{2}$ Regularization	0.01	0.01	0.01
Maximum epochs	50	50	50
Minimum batch size	20	20	20
Shuffle	every epoch	every epoch	every epoch

**Table 8 bioengineering-09-00529-t008:** Performance metrics of the test data from networks trained with the 35-mer adenosine dataset.

		ACC						MF1					
**Dataset**	**Hidden Units**	**Adam**		**RMSPROP**		**SGDM**		**Adam**		**RMSPROP**		**SGDM**	
		**ULSTM**	**BLSTM**	**ULSTM**	**BLSTM**	**ULSTM**	**BLSTM**	**ULSTM**	**BLSTM**	**ULSTM**	**BLSTM**	**ULSTM**	**BLSTM**
**Original Data**	**50**	0.333	0.333	0.167	**0.500**	0.333	0.333	0.500	0.500	0.500	**0.722**	0.583	0.500
	**100**	0.333	0.333	0.333	0.333	0.333	0.333	0.500	0.500	0.500	0.500	0.533	0.583
	**150**	0.333	**0.500**	**0.500**	0.333	0.333	0.333	0.533	**0.722**	**0.722**	0.500	0.533	0.583
	**200**	0.333	0.333	0.333	0.333	0.333	0.333	**0.583**	0.500	0.533	0.500	0.500	0.500
	**250**	0.333	0.333	0.333	0.333	**0.500**	0.333	0.500	0.500	0.533	0.500	**0.722**	0.500
	**300**	0.333	0.333	0.333	0.333	0.333	**0.500**	0.500	0.583	0.533	0.500	0.533	**0.722**
	**350**	0.333	0.333	0.333	**0.500**	**0.500**	0.333	0.500	0.500	0.500	**0.722**	**0.722**	0.500
	**400**	0.333	0.333	0.333	**0.500**	0.333	0.333	0.533	0.500	0.533	**0.722**	0.533	0.500
	**450**	0.333	0.333	0.333	0.333	0.333	0.333	0.533	0.500	0.533	0.500	0.533	0.500
	**500**	0.333	0.333	0.333	0.333	0.333	0.333	0.533	0.500	0.667	0.500	0.533	0.500
**50 Segments**	**50**	0.783	0.800	**0.883**	**0.867**	**0.450**	**0.400**	0.864	0.757	**0.886**	0.866	0.498	**0.523**
	**100**	0.750	0.883	0.850	0.817	0.350	0.383	0.707	0.861	0.803	0.773	0.419	0.393
	**150**	0.733	0.900	0.667	0.850	0.333	0.350	0.683	0.896	0.833	0.844	**0.544**	0.430
	**200**	0.783	0.833	0.633	0.800	0.350	0.350	0.769	0.804	0.681	0.794	0.411	0.431
	**250**	0.850	0.817	0.767	0.800	0.350	0.350	0.842	0.814	0.851	0.791	0.415	0.426
	**300**	0.850	0.800	0.767	0.850	0.350	0.350	0.842	0.763	0.736	0.837	0.437	0.432
	**350**	0.750	**0.900**	0.750	0.800	0.350	0.367	0.820	**0.900**	0.704	0.789	0.432	0.356
	**400**	0.850	0.850	0.817	0.550	0.350	0.367	0.803	0.844	0.808	0.673	0.418	0.362
	**450**	0.533	0.767	0.783	0.700	0.350	0.350	0.642	0.755	0.764	0.650	0.428	0.419
	**500**	**0.883**	0.800	0.333	0.817	0.367	0.350	**0.885**	0.794	0.549	**0.911**	0.367	0.419
**100 Segments**	**50**	**0.850**	0.825	0.833	0.875	0.542	0.458	**0.844**	0.817	0.828	0.869	0.530	0.450
	**100**	0.817	0.867	0.833	0.875	0.642	0.408	0.810	0.868	0.826	0.872	0.587	0.440
	**150**	0.825	0.917	**0.858**	0.917	0.642	0.525	0.817	0.913	**0.853**	0.914	0.591	0.453
	**200**	0.567	0.858	0.817	**0.925**	0.625	0.533	0.523	0.859	0.806	**0.924**	0.564	0.553
	**250**	0.850	0.883	0.842	0.842	0.642	0.625	0.843	0.881	0.836	0.843	0.602	0.566
	**300**	0.850	0.900	0.642	0.825	0.675	0.617	0.848	0.898	0.701	0.815	0.641	**0.652**
	**350**	0.817	0.850	0.492	0.817	0.692	**0.650**	0.809	0.845	0.713	0.804	0.670	0.597
	**400**	0.850	**0.933**	0.850	0.908	**0.700**	0.642	0.846	**0.932**	0.843	0.909	0.677	0.593
	**450**	0.825	0.908	0.750	0.792	**0.700**	0.625	0.813	0.906	0.708	0.784	**0.680**	0.568
	**500**	0.700	0.817	0.842	0.758	**0.700**	0.625	0.759	0.807	0.835	0.729	0.675	0.567

**Table 9 bioengineering-09-00529-t009:** Performance metrics of the OD test (control) data from the networks trained with the 35-mer adenosine dataset.

		ACC						MF1					
**Dataset**	**Hidden Units**	**Adam**		**RMSPROP**		**SGDM**		**Adam**		**RMSPROP**		**SGDM**	
		**ULSTM**	**BLSTM**	**ULSTM**	**BLSTM**	**ULSTM**	**BLSTM**	**ULSTM**	**BLSTM**	**ULSTM**	**BLSTM**	**ULSTM**	**BLSTM**
**Original Data**	**50**	0.462	0.462	0.385	0.462	0.308	0.385	0.679	0.679	0.471	0.701	0.431	0.610
	**100**	0.462	0.462	0.462	0.462	0.308	0.462	0.679	0.679	0.679	0.679	0.431	**0.729**
	**150**	0.385	0.462	0.462	0.462	0.308	**0.615**	0.564	**0.701**	**0.701**	0.679	0.414	0.726
	**200**	0.462	0.462	0.308	0.462	**0.385**	0.462	**0.701**	0.679	0.431	0.679	**0.564**	0.679
	**250**	0.385	0.462	0.308	0.538	0.308	0.462	0.564	0.679	0.414	0.634	0.473	0.679
	**300**	0.462	0.462	0.308	0.462	0.308	0.538	0.679	0.679	0.414	0.679	0.414	0.619
	**350**	0.462	0.462	0.385	**0.615**	**0.385**	0.462	0.679	0.679	0.583	**0.738**	0.439	0.679
	**400**	0.308	0.462	0.385	**0.615**	0.308	0.462	0.414	0.679	0.564	0.629	0.414	0.679
	**450**	0.308	0.462	0.308	0.462	0.308	0.462	0.414	0.679	0.414	0.679	0.414	0.679
	**500**	0.308	0.462	**0.538**	0.462	0.308	0.462	0.414	0.679	0.634	0.679	0.414	0.679
**50 Segments**	**50**	0.462	0.769	0.538	0.692	0.385	0.462	0.606	**0.886**	0.613	0.779	0.610	0.679
	**100**	0.538	0.538	0.538	0.692	0.462	0.462	0.731	0.707	0.707	0.817	0.679	0.679
	**150**	0.538	0.462	0.462	0.692	0.462	0.462	0.738	0.621	0.610	0.714	0.679	0.679
	**200**	0.615	0.769	0.462	**0.846**	0.462	0.462	0.711	0.805	0.563	**0.903**	0.679	0.679
	**250**	0.538	**0.769**	0.385	0.692	0.462	0.462	0.585	0.803	0.537	0.698	0.679	0.679
	**300**	0.615	0.692	0.385	0.615	0.462	0.462	0.673	0.821	0.537	0.697	**0.701**	0.679
	**350**	0.538	0.692	0.462	0.692	0.462	0.462	0.606	0.817	0.756	0.869	**0.701**	0.679
	**400**	0.615	0.769	0.538	0.615	0.462	0.462	0.704	0.776	0.631	0.738	0.679	0.679
	**450**	0.308	0.692	**0.692**	0.462	0.462	0.462	0.450	0.665	**0.781**	0.729	**0.701**	0.679
	**500**	**0.692**	0.615	0.385	0.538	0.462	0.462	**0.811**	0.771	0.564	0.617	**0.701**	0.679
**100 Segments**	**50**	0.615	0.538	0.462	0.692	0.308	0.462	0.605	0.638	0.513	0.779	0.450	0.729
	**100**	0.538	0.385	0.615	0.692	0.308	0.462	0.643	0.494	0.651	0.698	0.473	0.701
	**150**	**0.692**	0.615	**0.769**	0.769	0.308	0.462	0.725	0.730	0.754	0.785	0.500	0.701
	**200**	0.385	0.846	0.615	0.692	0.385	0.462	0.471	0.891	0.689	0.730	0.500	0.701
	**250**	0.615	0.769	0.692	0.538	0.385	0.462	0.714	0.810	**0.811**	0.607	0.500	0.729
	**300**	0.538	0.769	0.385	0.462	0.308	0.462	0.613	0.785	0.515	0.606	0.500	0.729
	**350**	**0.692**	0.692	0.308	**0.846**	0.385	0.462	**0.800**	0.817	0.467	**0.871**	0.500	0.729
	**400**	0.538	**0.923**	0.615	0.615	0.385	0.462	0.606	**0.943**	0.714	0.714	0.500	0.729
	**450**	0.538	**0.923**	0.615	0.538	0.385	0.462	0.648	**0.943**	0.605	0.617	0.500	0.729
	**500**	0.385	0.846	0.769	0.538	0.385	0.462	0.515	0.891	0.776	0.579	0.500	0.729

**Table 10 bioengineering-09-00529-t010:** Performance metrics of the test data from the networks trained with the 31-mer Oestrdiol dataset.

		ACC						MF1					
**Dataset**	**Hidden Units**	**Adam**		**RMSPROP**		**SGDM**		**Adam**		**RMSPROP**		**SGDM**	
		**ULSTM**	**BLSTM**	**ULSTM**	**BLSTM**	**ULSTM**	**BLSTM**	**ULSTM**	**BLSTM**	**ULSTM**	**BLSTM**	**ULSTM**	**BLSTM**
**Original Data**	**50**	0.333	0.333	**0.500**	0.500	0.333	0.333	0.533	0.533	0.778	**0.833**	0.667	0.533
	**100**	0.167	0.333	0.333	0.500	0.333	0.333	0.667	0.533	0.500	**0.833**	0.750	0.533
	**150**	**0.500**	0.500	0.333	0.500	0.333	0.167	0.722	**0.833**	0.533	0.722	0.533	0.400
	**200**	0.333	0.333	0.333	0.500	0.333	0.333	0.667	0.583	0.583	0.800	0.700	0.583
	**250**	**0.500**	0.500	0.333	0.500	0.333	0.333	0.722	0.722	0.667	0.722	0.700	0.533
	**300**	0.333	0.500	0.333	0.333	**0.500**	0.333	0.500	**0.833**	0.667	0.533	**0.800**	0.533
	**350**	0.333	0.333	0.333	0.333	0.333	0.333	**0.750**	0.583	0.700	0.533	0.700	0.533
	**400**	0.333	0.500	**0.500**	0.500	0.167	0.333	0.667	0.667	0.778	0.800	0.333	0.533
	**450**	0.333	0.500	0.333	0.333	0.167	**0.500**	0.500	0.722	0.700	0.533	0.500	**0.833**
	**500**	0.333	0.500	**0.500**	0.333	0.333	0.333	0.533	0.722	**0.800**	0.533	0.667	0.533
**50 Segments**	**50**	0.617	0.833	0.600	0.833	0.333	0.350	0.822	0.933	0.640	0.839	0.533	0.457
	**100**	0.617	0.900	0.717	0.800	**0.383**	0.367	0.665	0.887	0.685	0.888	0.518	0.496
	**150**	**0.833**	0.833	0.550	0.800	0.333	**0.400**	**0.933**	**0.933**	0.663	0.888	0.533	0.442
	**200**	0.617	0.833	0.600	0.800	0.333	0.350	0.754	**0.933**	0.608	0.893	**0.537**	0.421
	**250**	0.733	0.883	**0.750**	**0.833**	0.333	0.383	0.818	0.864	0.815	**0.933**	0.533	0.415
	**300**	0.583	**0.917**	0.517	0.717	0.333	0.350	0.748	0.909	0.630	0.786	0.533	0.432
	**350**	0.550	0.867	0.717	0.767	0.350	0.333	0.667	0.841	0.794	0.756	0.425	**0.530**
	**400**	0.500	0.833	**0.750**	**0.833**	0.350	0.367	0.722	**0.933**	**0.822**	0.920	0.425	0.364
	**450**	0.550	0.833	0.333	0.717	0.333	0.367	0.667	**0.933**	0.570	0.826	0.533	0.478
	**500**	0.700	0.483	0.167	0.333	0.350	0.333	0.777	0.598	0.290	0.551	0.425	0.530
**100 Segments**	**50**	0.708	0.867	0.667	**0.867**	0.633	0.558	0.790	0.835	0.746	0.833	**0.833**	**0.683**
	**100**	**0.800**	0.883	0.783	0.833	0.683	0.533	**0.876**	0.867	0.859	**0.921**	0.760	0.564
	**150**	0.708	0.867	0.708	0.842	**0.700**	0.525	0.780	0.839	0.794	0.798	0.783	0.553
	**200**	0.658	0.867	**0.800**	**0.867**	0.642	0.575	0.711	0.835	**0.880**	0.834	0.698	0.625
	**250**	0.700	0.883	0.792	**0.867**	0.608	**0.650**	0.779	0.864	0.747	0.836	0.654	0.610
	**300**	0.642	0.833	0.717	0.858	0.633	0.592	0.633	0.801	0.800	0.820	0.685	0.539
	**350**	0.758	0.900	0.658	**0.867**	0.600	0.617	0.834	0.893	0.730	0.835	0.641	0.578
	**400**	0.642	**0.900**	0.333	0.808	0.658	0.642	0.690	0.890	0.542	0.752	0.727	0.604
	**450**	0.708	0.867	0.792	0.858	0.608	0.608	0.790	0.842	0.870	0.820	0.654	0.556
	**500**	0.767	0.833	0.792	0.858	0.642	0.642	0.848	**0.921**	0.753	0.821	0.706	0.607

**Table 11 bioengineering-09-00529-t011:** Performance metrics of the OD test (control) data from the networks trained with the 31-mer Oestrdiol dataset.

		ACC						MF1					
**Dataset**	**Hidden Units**	**Adam**		**RMSPROP**		**SGDM**		**Adam**		**RMSPROP**		**SGDM**	
		**ULSTM**	**BLSTM**	**ULSTM**	**BLSTM**	**ULSTM**	**BLSTM**	**ULSTM**	**BLSTM**	**ULSTM**	**BLSTM**	**ULSTM**	**BLSTM**
**Original Data**	**50**	0.333	0.333	0.444	0.389	0.333	0.333	0.533	0.533	0.694	0.583	0.605	0.533
	**100**	0.389	0.333	0.333	0.556	0.333	0.333	0.519	0.533	0.503	0.708	**0.750**	0.533
	**150**	0.333	**0.556**	0.333	0.444	0.222	0.278	0.503	**0.708**	0.605	0.619	0.319	0.417
	**200**	0.333	0.444	0.278	**0.611**	**0.389**	0.333	0.605	0.656	0.500	**0.815**	0.625	0.583
	**250**	**0.500**	**0.556**	0.333	0.500	0.333	0.333	**0.722**	0.656	0.676	0.690	0.629	0.533
	**300**	0.333	0.389	0.333	0.333	0.333	0.389	0.523	0.528	0.605	0.533	0.605	0.550
	**350**	0.389	0.333	0.333	0.389	0.167	0.333	0.667	0.606	0.605	0.522	0.353	0.548
	**400**	0.333	0.444	**0.556**	0.500	0.167	0.333	0.605	0.595	0.693	0.800	0.353	0.533
	**450**	0.278	0.500	0.333	0.333	0.167	**0.500**	0.439	0.682	**0.700**	0.533	0.500	**0.833**
	**500**	0.222	0.500	0.333	0.333	0.333	0.333	0.319	0.690	0.605	0.533	0.605	0.533
**50 Segments**	**50**	0.667	0.722	0.611	0.611	0.333	0.333	**0.875**	0.796	0.765	0.597	0.533	**0.564**
	**100**	0.611	0.778	**0.778**	0.611	0.389	0.389	0.789	0.864	**0.855**	0.678	0.532	0.543
	**150**	**0.778**	0.778	0.500	0.556	0.333	0.389	0.871	0.864	0.690	0.597	**0.548**	0.532
	**200**	0.722	0.778	0.556	0.722	0.333	0.333	0.771	0.871	0.700	0.800	0.533	**0.564**
	**250**	0.722	0.778	0.667	0.778	0.333	0.389	0.800	**0.871**	0.731	**0.864**	0.533	0.543
	**300**	0.500	**0.833**	0.611	0.500	0.333	0.333	0.702	0.819	0.664	0.573	0.533	0.548
	**350**	0.556	0.722	0.722	0.667	0.333	0.389	0.678	0.810	0.764	0.752	**0.548**	0.543
	**400**	0.611	0.722	0.667	**0.778**	0.333	0.389	0.789	0.800	0.736	0.864	**0.548**	0.532
	**450**	0.500	0.667	0.333	0.667	0.333	0.333	0.690	0.714	0.605	0.780	0.533	0.564
	**500**	0.667	0.500	0.167	0.333	**0.389**	**0.389**	0.731	0.821	0.286	0.564	0.532	0.543
**100 Segments**	**50**	0.611	0.778	0.722	0.778	0.667	0.444	0.836	0.864	0.800	0.864	0.875	0.656
	**100**	**0.833**	0.722	**0.833**	0.722	0.667	0.500	**0.933**	0.817	**0.921**	0.796	0.731	0.649
	**150**	**0.833**	0.778	0.722	0.722	0.667	0.556	0.921	**0.900**	0.788	0.796	**0.875**	0.599
	**200**	0.667	0.778	0.778	0.778	0.611	0.611	0.875	0.864	0.871	0.864	0.801	0.681
	**250**	0.667	0.778	0.778	0.778	0.611	0.611	0.851	0.758	0.886	0.864	0.789	0.681
	**300**	0.667	0.778	0.722	0.778	0.611	0.611	0.767	0.770	0.800	**0.871**	0.789	0.681
	**350**	0.778	0.722	0.722	0.778	0.611	0.611	0.853	0.717	0.788	**0.871**	0.789	0.681
	**400**	0.667	0.778	0.278	0.722	0.667	0.611	0.875	0.864	0.486	0.715	0.739	0.681
	**450**	0.722	0.778	0.778	0.778	0.611	0.611	0.809	0.864	0.871	**0.871**	0.789	0.681
	**500**	0.722	0.778	0.778	0.778	0.611	0.611	0.800	0.900	0.766	0.864	0.789	0.681

**Table 12 bioengineering-09-00529-t012:** Performance metrics of the test data from the networks trained with the 35-mer Oestrdiol dataset.

		ACC						MF1					
**Dataset**	**Hidden Units**	**Adam**		**RMSPROP**		**SGDM**		**Adam**		**RMSPROP**		**SGDM**	
		**ULSTM**	**BLSTM**	**ULSTM**	**BLSTM**	**ULSTM**	**BLSTM**	**ULSTM**	**BLSTM**	**ULSTM**	**BLSTM**	**ULSTM**	**BLSTM**
**Original Data**	**50**	0.300	0.400	**0.400**	**0.500**	0.200	**0.500**	0.450	0.546	**0.579**	**0.667**	0.500	**0.656**
	**100**	0.200	0.400	0.200	0.400	**0.400**	0.300	0.343	0.567	0.343	0.579	**0.583**	0.395
	**150**	0.300	0.400	0.300	0.400	0.300	0.300	0.486	0.533	0.486	0.546	0.500	0.429
	**200**	0.400	0.400	0.300	0.400	0.200	0.300	0.546	0.533	0.486	0.546	0.500	0.429
	**250**	0.300	**0.500**	0.300	0.300	0.300	0.300	0.536	**0.689**	0.450	0.619	0.500	0.429
	**300**	0.400	**0.500**	0.300	**0.500**	0.200	0.300	0.556	0.592	0.433	0.592	0.500	0.429
	**350**	0.400	**0.500**	0.200	0.400	0.200	**0.500**	**0.579**	0.592	0.367	0.546	0.500	0.635
	**400**	**0.500**	0.400	0.300	**0.500**	**0.400**	0.300	0.547	0.546	0.450	0.600	**0.583**	0.429
	**450**	0.400	**0.500**	0.300	0.400	**0.400**	0.400	0.522	0.592	0.450	0.546	**0.583**	0.544
	**500**	0.300	0.400	0.300	0.400	**0.400**	0.300	0.450	0.546	0.450	0.546	**0.583**	0.429
**50 Segments**	**50**	0.617	0.617	0.617	0.617	0.367	0.433	0.646	0.660	0.665	0.668	0.459	0.511
	**100**	0.667	0.650	**0.667**	0.650	0.333	0.483	0.729	0.706	0.725	0.629	0.508	0.547
	**150**	0.683	0.767	**0.667**	**0.667**	0.333	0.483	0.747	0.747	**0.729**	0.640	0.513	0.553
	**200**	0.683	0.700	0.583	0.617	0.333	0.417	0.747	0.685	0.636	0.640	0.513	0.480
	**250**	0.683	0.667	0.617	**0.667**	**0.383**	**0.500**	0.647	0.624	0.579	**0.722**	0.503	0.600
	**300**	**0.750**	0.733	0.650	**0.667**	0.333	0.417	**0.748**	0.724	0.690	0.707	0.518	0.508
	**350**	0.617	0.750	0.600	**0.667**	0.333	0.450	0.677	0.731	0.665	0.722	**0.580**	0.509
	**400**	0.700	0.733	0.417	0.517	0.333	0.450	0.696	0.723	0.559	0.545	0.530	**0.656**
	**450**	0.633	0.567	0.333	0.633	0.333	0.500	0.588	0.733	0.512	0.628	0.508	0.564
	**500**	0.567	**0.833**	0.633	0.550	0.333	0.450	0.614	**0.836**	0.690	0.517	0.542	0.627
**100 Segments**	**50**	0.633	0.742	0.683	0.750	0.600	0.500	0.632	0.718	0.621	0.733	0.611	0.480
	**100**	0.742	0.900	0.692	0.783	0.492	0.525	0.726	0.899	0.630	0.779	0.489	0.638
	**150**	0.767	0.867	0.683	0.783	0.525	0.533	0.753	0.862	0.605	0.783	0.526	0.530
	**200**	0.775	0.842	0.725	**0.892**	0.592	0.550	0.737	0.841	0.695	**0.892**	0.630	0.557
	**250**	0.792	0.867	0.725	0.858	0.492	0.525	0.774	0.865	0.699	0.857	0.612	0.541
	**300**	0.800	0.850	**0.750**	0.808	0.617	0.600	0.784	0.845	0.694	0.797	0.657	**0.644**
	**350**	0.742	**0.933**	0.583	0.800	0.533	0.567	0.722	**0.933**	**0.709**	0.778	0.549	0.600
	**400**	0.575	0.917	0.600	0.683	**0.642**	0.600	0.573	0.916	0.604	0.735	**0.698**	0.640
	**450**	**0.825**	0.783	0.600	0.650	0.608	**0.633**	**0.801**	0.777	0.643	0.699	0.635	0.580
	**500**	0.617	0.858	0.600	0.783	0.592	0.592	0.614	0.858	0.593	0.775	0.621	0.532

**Table 13 bioengineering-09-00529-t013:** Performance metrics of the OD test (control) data from the networks trained with the 35-mer Oestrdiol dataset.

		ACC						MF1					
**Dataset**	**Hidden Units**	**Adam**		**RMSPROP**		**SGDM**		**Adam**		**RMSPROP**		**SGDM**	
		**ULSTM**	**BLSTM**	**ULSTM**	**BLSTM**	**ULSTM**	**BLSTM**	**ULSTM**	**BLSTM**	**ULSTM**	**BLSTM**	**ULSTM**	**BLSTM**
**Original Data**	**50**	0.350	0.450	**0.350**	0.450	0.200	0.300	0.517	0.613	0.469	0.596	0.267	0.462
	**100**	0.300	0.400	0.200	0.350	**0.300**	0.300	0.472	0.548	0.393	0.468	0.333	0.445
	**150**	0.350	0.400	0.350	0.450	0.200	0.350	0.472	0.522	0.511	0.613	**0.500**	**0.522**
	**200**	**0.450**	0.400	**0.350**	0.400	0.150	0.350	**0.607**	0.543	0.456	0.538	0.353	0.500
	**250**	0.400	0.400	0.300	0.350	0.250	0.350	0.537	0.543	**0.600**	0.531	0.347	**0.522**
	**300**	0.300	0.450	**0.350**	0.400	0.150	0.350	0.400	0.632	0.420	0.604	0.333	0.500
	**350**	0.250	0.450	**0.350**	0.450	0.200	0.350	0.422	0.539	0.532	0.613	0.271	0.508
	**400**	0.200	0.450	0.250	0.450	0.150	0.350	0.301	0.638	0.426	0.627	0.300	0.508
	**450**	0.300	0.450	0.250	0.450	0.250	0.250	0.489	0.671	0.550	**0.639**	0.322	0.298
	**500**	0.250	0.450	0.250	0.450	**0.300**	0.350	0.588	**0.639**	0.453	0.627	0.383	0.500
**50 Segments**	**50**	0.650	0.550	0.600	0.550	0.300	0.400	**0.756**	0.644	0.687	0.627	0.431	**0.598**
	**100**	0.600	0.500	0.650	**0.650**	0.300	0.400	0.705	0.568	0.740	0.635	0.472	0.561
	**150**	0.650	0.550	0.650	**0.650**	0.300	0.400	0.750	0.576	0.745	0.635	0.517	**0.598**
	**200**	0.650	**0.600**	**0.700**	0.550	**0.350**	0.400	0.750	0.622	**0.785**	**0.717**	0.444	0.578
	**250**	0.550	0.550	0.500	0.550	**0.350**	0.400	0.607	0.644	0.550	0.627	0.476	0.482
	**300**	0.550	0.500	0.600	0.550	0.300	0.400	0.608	0.558	0.673	0.644	0.500	**0.598**
	**350**	0.550	0.550	0.550	0.500	0.300	0.400	0.607	**0.644**	0.643	0.643	**0.558**	0.561
	**400**	**0.750**	0.550	0.350	0.600	0.300	0.400	0.749	0.593	0.447	0.690	0.517	0.561
	**450**	0.550	0.550	0.250	0.600	**0.350**	0.400	0.634	0.627	0.483	0.625	0.444	0.561
	**500**	0.600	**0.600**	0.600	0.350	0.300	0.400	0.700	0.638	0.688	0.539	0.500	0.561
**100 Segments**	**50**	0.600	0.550	0.550	0.550	0.600	0.450	0.665	0.576	0.653	0.576	0.675	0.594
	**100**	0.600	0.550	0.550	0.600	0.550	0.450	0.614	0.576	0.545	0.605	0.648	0.613
	**150**	0.600	**0.600**	0.550	0.550	0.500	0.450	0.613	0.638	0.542	0.670	0.577	0.613
	**200**	0.600	0.550	0.600	0.600	0.550	0.450	0.606	0.590	0.587	0.641	0.648	0.613
	**250**	**0.650**	**0.600**	0.600	**0.650**	0.500	**0.550**	**0.698**	0.638	0.593	**0.701**	**0.713**	**0.644**
	**300**	0.600	0.500	600	0.600	**0.600**	0.400	0.631	0.503	0.636	0.605	0.607	0.522
	**350**	0.550	0.550	0.550	0.550	0.500	0.450	0.593	0.538	**0.717**	0.630	0.697	0.613
	**400**	0.500	**0.600**	0.600	0.600	0.600	0.400	0.608	0.620	0.561	0.641	0.607	0.522
	**450**	0.650	0.500	**0.700**	0.600	0.600	0.400	0.655	**0.657**	0.691	0.665	0.695	0.522
	**500**	0.600	0.550	0.600	0.500	0.550	0.450	0.667	0.590	0.665	0.610	0.656	0.613

## Data Availability

The data presented in this study might be available on request from the corresponding author. There are restrictions on data availability due to their necessity for our future work.

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
