# Peer review of "Predicting Analyte Concentrations from Electrochemical Aptasensor Signals Using LSTM Recurrent Networks"

_bioengineering, 2022, doi:10.3390/bioengineering9100529_

Round 1

Reviewer 1 Report

A couple of minor typos to correct:

Table 2. caption. typo "complately"

Line 470 type "th e"

Also, even though it is mentioned on line 354 that the experiments were performed in Matlab Deep Learning Toolbox, this fact should probably be made more obvious up front in the methods section.

Author Response

Dear reviewer,

We would like to thank you for taking the necessary time and effort to review the manuscript. ‎We sincerely appreciate all your valuable comments and suggestions, which helped us in improving the ‎quality of the submitted manuscript.‎

The following items explain the applied changes according to your comments:

Point 1:  Table 2. caption. typo "complately".

Response 1: The typo was changed to “completely”.

Point 2:  Line 470 type "th e".

Response 2: The typo in line 470 was changed to “the”.

Point 3:  Also, even though it is mentioned on line 354 that the experiments were performed in Matlab Deep Learning Toolbox, this fact should probably be made more obvious up front in the methods section.

Response 3: Regarding this point, the following sentence was added to the LSTM Optimization subsection of the method section:‎

‎“It should be noted that all deep learning algorithms were implemented by MATLAB R2021b ‎Deep Learning Toolbox.”‎

On behalf of all the co-authors.

Yours sincerely,

Associate Professor Alan Wang

Auckland Bioengineering Institute, Faculty of Medical and Health Sciences

The University of Auckland, Auckland, New Zealand

Reviewer 2 Report

The authors have proposed a method for producing more synthetic data and explored LSTM networks to analyze the drain current signals. The LSTM networks are successful in predicting the analyte concentration. The analysis details are sound and the scientific content is meaningful for the interests of the readers. I can recommend the publication of this manuscript.

Author Response

Dear reviewer,

We would like to thank you for taking the necessary time and effort to review the manuscript. ‎

On behalf of all the co-authors.‎

Yours sincerely,

Associate Professor Alan Wang

Auckland Bioengineering Institute, Faculty of Medical and Health Sciences

The University of Auckland, Auckland, New Zealand
